# Vestibular Dysfunction and the Leftward Deviation in the New Line Bisection Task Using Three-Dimensionally Transformed Rectangles

**DOI:** 10.3390/audiolres15040086

**Published:** 2025-07-15

**Authors:** Teru Kamogashira, Shinnosuke Asakura, Hideaki Funayama, Kenji Ito, Noriaki Sunaga, Nao Shikanai, Fumihiko Itagaki, Toshitaka Kataoka, Shizuka Shoji, Megumi Koizumi, Shinichi Ishimoto

**Affiliations:** 1Department of Otolaryngology, JR Tokyo General Hospital, Tokyo 151-8528, Japan; 2Department of Otolaryngology and Head and Neck Surgery, Faculty of Medicine, University of Tokyo, Tokyo 113-8655, Japan; 3Department of Clinical Examination, JR Tokyo General Hospital, Tokyo 151-8528, Japan; 4Department of Artificial Organs Medical Device Creation, Tokyo Medical Center, Tokyo 169-0073, Japan; 5Faculty of Business Administration, Asia University, Tokyo 180-8629, Japan

**Keywords:** line bisection task, deviation, pseudoneglect, vestibular dysfunction, caloric testing, vHIT

## Abstract

**Background/Objectives**: The line bisection task (LBT) is a well-known test in which a horizontal line is presented in front of the subject and the subject is asked to draw a mark vertically bisecting the line. We developed the new LBT using three-dimensionally transformed rectangles to enhance the sense of depth and evaluated the influence of vestibular dysfunction on the deviation. **Methods**: One hundred participants were recruited from patients referred to the vertigo outpatient clinic. The average deviation in the LBT was the leftward deviation in the figures viewed from the right side and the rightward deviation in the figures viewed from the left side, indicating that the figures were perceived three-dimensionally, with the division point deviating to the far side. **Results**: In multivariate analysis of variance (MANOVA) analyses, the significant leftward deviation was observed in the group with vestibular dysfunction in caloric testing, and the significant rightward deviation dependent on increasing age was also observed. In univariate analyses, the significant leftward deviation in the figure viewed from the left side (135 degrees) was observed in the group with vestibular dysfunction in caloric testing, and the significant leftward deviation was also observed in figures viewed from the center, left or right side (0, 15 and 165 degrees) in the group with vestibular dysfunction in vHIT evaluation. **Conclusions**: Vestibular dysfunction can alter the deviation in the new LBT, suggesting the potential of the new LBT as an assessment of vestibular dysfunction.

## 1. Introduction

The line bisection task (LBT) is a well-known test in which a line is presented in front of the subject and the subject is asked to draw a mark vertically bisecting the line. The test was originally designed to detect visual field defects due to brain damage [1], and has been commonly used as a test for hemispheric neglect in clinical practice since around 1980. When this test is performed on normal subjects without brain damage, the division position tends to shift slightly to the left of the actual midpoint, which is called pseudoneglect (PN), and is considered to be one of the evidences of the dominance of the right cerebral hemisphere in information processing of the both hemispheres [2,3]. Many factors including age, sex, hemispheric dominance (dominant hand), direction of eye movements, line length, line direction, and variety of bisecting stimuli have been reported to influence the presence, degree, and direction of PN [2,3].

There are several reports of vestibular dysfunction in PN, with many studies referring to the function of neural circuits in the cerebrum. The hemispatial neglect can develop after acute unilateral peripheral vestibulopathy, which is attributed to damaged vestibular subnuclei, which receive afferents from both peripheral vestibular end organs and the vestibulocerebellum and project to the ipsilateral or contralateral thalamus and vestibular cortex [4]. However, in another study, vestibular tone imbalance due to unilateral impairment of the vestibular organs did not cause spatial hemineglect, with mild attention deficits in both visual spaces [5].

Visuospatial neglect involves the central influence of vestibular stimuli on the mechanisms of spatial representation [6]. The bias to the right suggests that the subject on the left is being ignored, which occurs in the superior temporal cortex, insula, and temporo-parietal junction involved in the multisensory system, including vestibular function. These areas integrate multimodal functions of vestibular, auditory, cervical proprioceptive, and visual input to form higher-order spatial representations [7].

Asymmetric vestibular input to cortical areas leads to representational spatial deficits and spatial cognition, as well as deficits in cortical processing of vestibular input in spatial neglect after right hemisphere stroke [8]. Several studies on rehabilitation for post-stroke hemispatial neglect showed that vestibular stimulation may have therapeutic potential; however, results have been inconsistent and further studies based on more careful methodology are needed [9].

The vestibular function can be altered by applying an electric current to the posterior part of the ear, which is called galvanic vestibular stimulation (GVS). The left or right vestibular imbalance caused by GVS can alter PN of the LBT to each side [10,11]. Ferré et al. reported that GVS induced polarity-dependent effects in spatial perception in normal subjects, suggesting that GVS may affect the direction-dependent perception of body orientation in space (left-anodal and right-cathodal GVS induced a leftward bisection bias, while right-anodal and left-cathodal GVS induced a rightward bisection bias) [11]. Rorsman et al. assessed patients with right hemisphere stroke with left neglect and found that left-anodal and right-cathodal GVS decreased left neglect, i.e., induced leftward deviation [12]. Utz et al. investigated the impact of GVS on performance in the LBT in right-brain-damaged patients with or without left-sided visual neglect, and found that both left-cathodal and right-cathodal GVS significantly reduced the rightward line bisection error (i.e., the leftward deviation compared to baseline and sham stimulation) as compared to the baseline and sham stimulation in left-neglect patients, but not in control patients [13]. Oppenländer et al. reported that in patients with unilateral right-sided stroke with or without left neglect, subliminal right-anodal and left-cathodal GVS induced leftward bisection bias compared to sham stimulation, implicating the possibility of neurorehabilitation with GVS [14]. GVS may produce lasting reductions in neglect that are clinically important [15]. Contrary to the above studies, Ruet et al. evaluated the effect of GVS on the deviation in the LBT in patients with a first right ischemic or hemorrhagic stroke and unilateral spatial neglect, and found that GVS did not reduce spatial neglect symptoms in any of the stimulation conditions [16]. Based on these previous studies, the LBT can be a valuable tool for the evaluation of vertigo or dizziness disorders including the vestibular function.

We developed the new LBT using three-dimensionally transformed rectangles to enhance the sense of depth. Several previous studies have reported the importance of depth perception in the LBT, and our new LBT uses deformation to represent depth. Depth perception is influenced by left–right alignment along the horizontal axis [17], and visuospatial reality construction through virtual reality was associated with a rightward LBT bias in the virtual environment [18]. The preference for rectangularity is stronger than that for symmetry [19], and rectangularity is effective for shape perception by incorporating some additional constraints [20]. The purpose of this study is to evaluate the influence of the vestibular function on the deviation in the new LBT with three-dimensionally transformed rectangles in cases with vestibular dysfunction.

## 2. Materials and Methods

### 2.1. Standard Protocol Approvals, Registrations, and Participant Consent

This clinical study was conducted in a single institution. One hundred participants were recruited from patients referred to the vertigo outpatient clinic, Department of Otolaryngology, JR Tokyo General Hospital, between August 2020 and August 2022. Inclusion criteria were as follows: (1) Male or female patients ≥ 20 years of age. (2) Written informed consent must be obtained before any study-related assessment is performed. Exclusion criteria were as follows: (1) Persons with visual impairment. (2) Those who have difficulty operating a computer. The participants of the control group were recruited from students of Asia University (only Table A7). The clinical study was approved by the regional ethical standards committee of JR Tokyo General Hospital (approval code: R02-03; approval date: 29 July 2020; trial registration number: University Hospital Medical Information Network (UMIN) Clinical Trials Registry (UMIN-CTR), UMIN000041573; date of registration: 29 July 2020). The control study was approved by the research ethics committee of Asia University.

### 2.2. Procedures

#### 2.2.1. New Line Bisecting Task (LBT)

The new LBT was performed using a computer with the application software installed (Python3/PsychoPy2 for Windows version 1.90.3 [21]) following the procedures below: (1) The inspector asks the examinee to place their chin on a stand 60 cm away from the 24-inch liquid crystal display (LCD) monitor to fix the face position. The edges of the monitor are captured in the peripheral vision and will not affect the measurement of the participant [22,23]. (2) The inspector asks the examinee to look at the cross-shaped figure displayed in the center of the LCD monitor and adjusts the position so that the examinee’s eye level and the center are aligned. (3) The inspector dims the lights in the inspection room. (4) The inspector asks the examinee to divide the quadrangle figures displayed on the LCD monitor in half horizontally, regardless of the shape or the area of the figure, with a movable vertical line using the mouse and click the button. The presented figures are rectangles and trapeziums, which are three-dimensionally equivalent to rectangles viewed from right-side or left-side perspectives and are projected onto a plane. The presented figures are always centered on the LCD. The view rotation angles are 0, 15, 30, 45, 60, 75, 105, 120, 135, 150 and 165 degrees (11 types) (Figure 1). The movable vertical line is displayed from the upper left, lower left, upper right or lower right outside of the displayed figure for each type (4 display starting positions). (5) The inspector conducts preliminary 4 exercises and starts the actual experiment if there are no questions from the examinee. (6) A total of 44 figures with 11 types and 4 display starting positions are displayed. (7) The inspector measures the time taken for the test. (8) The deviation from the true center of the figure is recorded and the average and standard deviation are calculated for all 11 types, respectively. The leftward deviation from the true center is shown in negative numbers, and the rightward deviation from the true center is shown in positive numbers.

#### 2.2.2. Vestibular Function Examination and Questionnaire

Caloric testing was performed in a darkened room by irrigating the external auditory canal with 20 mL ice water (4 °C) for 10 s. Caloric nystagmus was recorded using videonystagmography (Interacoustics, Middelfart, Denmark). An abnormal caloric response was defined by either of the following criteria: (1) left or right canal paresis (CP); CP percentage ≥ 20% [24]; or (2) both-sided CP; maximum slow-phase eye velocity < 10 degree/s [25].

The video head impulse test (vHIT) was performed using ICS Impulse 2.00 Build 605 Firmware 1.3 (Otometrics, Taastrup, Denmark) according to the instruction of the manufacturer. VOR gains were analyzed based on the manufacturer’s algorithms, using 175 samples out of a total of 250 samples obtained on each trial. Data from the onset of head motion and subsequent zero crossing of head velocity were used to measure the area under the curve (AUC) of head velocity. The value of VOR gain was calculated as (AUC of eye velocity)/(AUC of head velocity). The gain of <0.8 for the lateral semicircular canal or <0.7 for the anterior or posterior semicircular canal of the single side (left or right) or both sides was considered to be abnormal function evaluated with vHIT. Due to equipment availability, vHIT was performed in 40 cases.

The cervical vestibular-evoked myogenic potential (cVEMP) and ocular VEMP (oVEMP) were recorded with the Nicolet EDX system (Natus) using a tone burst stimulus of 500 Hz at 125 dBpeSPL (rise, 1 ms; plateau, 2 ms; fall, 1 ms) and a tone burst of 1 kHz at 125 dBpeSPL (rise, 1 ms; plateau, 2 ms; fall, 1 ms). In cVEMP testing, electromyographic (EMG) activity was recorded from a surface electrode placed on the upper half of each sternocleidomastoid muscle (SCM), with a reference electrode on the side of the upper sternum and a ground electrode on the chin. During the recording, in the supine position, subjects were instructed to raise their heads from the pillow in order to contract the SCM. The EMG signal from the stimulated side was amplified and bandpass-filtered (20–2000 Hz). The stimulation rate was 5.1 Hz, and the analysis time was 100 ms. Responses to 20 stimuli were average three times [26]. In oVEMP testing, subjects lay supine on a bed, with their head supported by a pillow and with surface EMG electrodes placed on the skin 1 cm below (active) and 3 cm below (indifferent) the center of each lower eyelid. The ground electrode was placed on the chin. During testing, the subject looked up approximately 30 degrees above straight ahead and maintained their focus on a small dot approximately 1 m from their eyes. The signals were amplified by a differential amplifier (bandwidth: 0.5–500 Hz). Responses to 20 stimuli were averaged three times [26]. An abnormal VEMP response was defined by the following criteria: asymmetry ratio (AR) percentage ≥ 33% or no response in both ears.

Questionnaire regarding psychological symptoms included Dizziness Handicap Inventory (DHI) [27], Self-rating Depression Scale (SDS) (age less than 65) [28] or Geriatric Depression Scale (GDS) (age 65 or over) [29,30], Hospital Anxiety and Depression Scale (HADS) [31], POUNDing (Pulsating, duration of 4–72 hOurs, Unilateral, Nausea, Disabling) [32], Migraine Disability Assessment (MIDAS) [33] and 4-item migraine screener (headache exacerbation in daily performance, nausea, light sensitivity, and hypersensitivity to odors) [34].

### 2.3. Data Analysis

Excel for Microsoft 365 (version 2207, Microsoft, Redmond, WA, USA) was used for processing data. LBT analysis and statistical analysis were performed using R version 4.3.3 software (R Core Team; R Foundation for Statistical Computing, Vienna, Austria, 2024) [35] with tableone package version 0.13.2. Data are expressed as mean (standard deviation). Multivariate analysis of variance (MANOVA) was used to evaluate dependent variables of deviations after confirming normality using the Kolmogorov–Smirnov normality test (*p* > 0.05) and equality of variances using Levene’s test (*p* > 0.05). Student’s t test was used to compare the average between two groups and analysis of variance (ANOVA) was used to compare between more than two groups. The Pearson product–moment correlation coefficient was calculated to analyze correlations between parameters. Regression coefficients in multiple regression were calculated to analyze dependent variables. *p* < 0.05 was considered statistically significant.

## 3. Results

Characteristics of the participants are shown in Table 1 and Table A1. There were four left-handers. Three cases had a history of cerebellar infarction. There were no cases of dementia. A positive value means that the division position is to the right of the true center of the figure, and a negative value means a position to the left (Figure 1). The average deviation in the LBT was leftward deviation in the figures viewed from the right side and rightward deviation in the figures viewed from the left side, indicating that the figures were perceived three-dimensionally, with the division point deviating to the far side. The deviation in the figure viewed from the center showed no deviation (Table A2). The distributions of deviation are shown in Figure A1.

Significant leftward deviation was observed in vestibular dysfunction. In multivariate analysis of variance (MANOVA) analyses, significant leftward deviation was observed in the group with vestibular dysfunction in caloric testing, and significant rightward deviation dependent on increasing age was also observed (Table 2). In univariate analysis, there were significant correlations between deviations and age in figures of 0, 15, 45, 60 and 75 degrees (Figure A2, Table 3), and the deviation in the LBT of the figure viewed from the left side (135 degrees) showed significant leftward deviation in the group with CP compared to the group without CP, whereas there was no significant difference in other figures (Table 4). Leftward deviation was observed in both left and right CP compared to cases without CP, whereas there was no significant difference between left and right CP (Table 5). There was no significant difference in the deviation between males and females (Table A2). In MANOVA analyses, significant age-dependent deviation was observed but not in vHIT (Table A3). In univariate analyses, significant leftward deviation in the abnormal group in lateral vHIT was also observed in figures viewed from the center, left or right (0, 15 and 165 degrees) (Table A4). Leftward deviation was observed in left, right and both abnormal cases in lateral vHIT compared to normal cases in lateral vHIT, whereas there was no significant difference between left, right and both abnormal cases in lateral vHIT (Table A5). There was no significant difference in the deviations between the group with normal AR of VEMP and with abnormal AR of VEMP (Table A6).

There was no significant correlation between the deviations of the LBT and psychological scores (Table 6). Among the deviations of the LBT, there were high correlations between figures with close angles. The questionnaire scores showed high correlations between HADS-A and GDS (0.64), and between HADS-D and GDS (0.66), while other parameters showed low correlations.

Regression coefficients in multiple regression between the deviation in 135 degrees and age or CP showed significance only in CP (age: *p* = 0.9023; CP: *p* = 0.0358 *). Regression coefficients in multiple regression between the deviation in 0, 15 and 165 degrees and age or abnormal lateral vHIT showed significance in both age and vHIT in 0 degrees (age: *p* = 0.0394 *; vHIT: *p* = 0.0293 *), significance in age in 15 degrees (age: *p* = 0.0448 *; vHIT: *p* = 0.0808), and no significance in 165 degrees (age: *p* = 0.0677; vHIT: *p* = 0.0697).

The deviation in the figure viewed from the center showed significant leftward deviation in the control group of Asia University (Table A7). As noted above, age was found to significantly affect deviation, so direct comparisons between the control group of Asia University and the outpatient group were not made because the age distribution was very different from that of outpatient cases.

## 4. Discussion

In this study, PN, which is identical to leftward deviation, was not observed in the total average of any figures. However, the vestibular dysfunction partly affected the leftward deviation. The directions of deviations under the left or right vestibular dysfunction were both left-sided compared to normal subjects, which was identical to the deviation side of PN, suggesting the relevance of the LBT to PN as found in in previous reports.

There was no significant difference in the deviation between male and female, whereas there were significant positive correlations between deviation and age in several figures viewed from the right side. Interestingly, the leftward deviation in figures from the right side decreased with age, so aging may lead to rightward deviation. This age-related rightward shift was also observed in several studies [36,37]. Age-related changes could influence the brain mechanisms of visuospatial attention, affecting the deviation [3]. The rightward shift with increasing age (Table 2 and Table 3) is consistent with a significant leftward shift of the figure viewed from the center only in the control group of Asia University but not in the outpatient group with older people. The lack of a significant left bias in the outpatient group may have been influenced by a rightward bias due to age.

The effects of GVS on the line bisection deviation varied across studies; however, the deviations tended to be biased to the left under GVS relative to baseline. Although a few subjects in this study have cerebellar infarction without hemisphere strokes, some studies on patients with brain lesions also observed leftward deviation under GVS. In previous studies, GVS basically induced leftward deviation in the LBT. In most cases of vestibular dysfunction, the affected side is unilateral, which may be similar to the unbalanced condition under GVS. Deviation to the left side was observed in both left and right vestibular dysfunction cases, both in caloric testing and in vHIT evaluation, but the number of cases was too small to compare the deviation between left and right vestibular dysfunction.

Another mechanism of the leftward deviation under GVS could be that GVS increased intrinsic alertness. Manly et al. reported that sleep deprivation and session continuation caused a significant rightward shift [38]. Matthias also reported similar results that low intrinsic alertness can cause left-side-neglect-like performance with rightward deviation [39]. However, Smaczny reported that alertness had no effect on spatial attention as measured by the LBT [40]. Input from the vestibular system may be stressful in the brain system and exacerbate alertness, so vestibular dysfunction may lead to the reduction of information input and may conversely improve alertness in the brain system. Further evaluation of vestibular function and alertness is needed.

The vestibular dysfunction can result in a predominance of visual information [41,42], which may influence the PN phenomenon. The vestibular system and its hemispheric dominance can mature early during ontogenesis and are localized in opposite hemispheres [43]. DNA methylation in the DBH (dopamine beta-hydroxylase) promoter region and other dopamine-related genes affected line bisection deviation in right-aligned trials, suggesting the influence of hemisphere dominance type and attentional bias on the line bisection deviation [44]. Vestibular dysfunction of left, right or both sides and vestibular imbalance may lead to an increase in the weight of visual information, which causes PN (i.e., leftward deviation) in the LBT.

The deviation in the vHIT evaluation showed significant differences at 0, 15 and 165 degrees, and regression coefficients in multiple regression showed significance at 0 and 15 degrees in the vHIT evaluation, so the widths of figures of 75 and 105 degrees may be too small to sufficiently detect significant deviation, which necessitates further analysis of the effect of the figure width because the deviation was affected by the three-dimensional perception. The sensitivity of the vestibular dysfunction is higher in caloric testing than in vHIT, and the specificity is higher in vHIT than in caloric testing [45,46]. More figures (three figures) showed leftward deviations in vestibular dysfunction in vHIT than in caloric testing (one figure), suggesting that leftward deviations tend to occur under more severe vestibular dysfunction.

No significant correlation was found between the task time or deviations in the LBT and questionnaire scores regarding psychological symptoms in this study. One study of stroke patients found a significant association between the LBT time and prognosis of hemispatial neglect, indicating that cognitive deficits such as attention and visual perception affect recovery from hemispatial neglect [47]. In a study examining the LBT in normal subjects, generalized anxiety disorder (GAD), and treatment-resistant depression (TRD) outpatients, a significant leftward deviation was observed in TRD patients compared to normal subjects [48]. In another study, patients with dependent personality disorder bisected significantly leftward compared to healthy controls [49]. In an LBT study on headache, normal subjects and migraine patients showed rightward deviation, while tension-type headache patients, in whom psychological stress is mainly involved, showed leftward deviation compared to normal subjects and migraine patients, suggesting that right hemisphere activation may be relatively stronger or left hemisphere activation weaker in tension-type headache [50]. While the control group of Asia University showed a significant leftward deviation at 0 degrees, the outpatients in this study showed no significant deviation at 0 degrees, which could be considered the opposite results of previous studies above, given that the patients tended to show more depression than the control group of Asia University. The age factor may contribute more strongly to the deviation in the LBT than the psychogenic factor. In addition, a leftward deviation was observed in vestibular dysfunction regardless of left-side or right-side perspectives, suggesting that the impaired side of the vestibule was less affected.

In terms of approach motivation, significant rightward deviation was observed in challenge conditions with high self-esteem, indicating the effectiveness of LBT in measuring behavioral neuroscience indicators [51]. In another study, approach-motivated individuals exhibited a significant rightward deviation compared to avoidance-motivated individuals, but only under conditions of high time pressure [52]. However, other research groups have argued that the LBT was not suitable for measuring lateralized approach/avoidance biases, probably because of the large individual differences inherent in visuospatial bias [53].

There are several limitations in this study. Firstly, an analysis of the dominant hand was not conducted because the number of cases with left-handers was small. Secondly, the chronological changes of the deviation were not evaluated, and the vestibular functions may change depending on the physical condition and climate in some vestibular disorders. SVV testing was not performed, and additional evaluation could be considered. Thirdly, the number of cases is insufficient to apply this new LBT clinically, because there are multiple tests to assess vestibular functions including caloric testing, vHIT and VEMP. Although vestibular disorders increase with age, the number of cases of vestibular disorders at younger ages is small, making analysis of the interaction difficult. Fourthly, cases in this study included a diverse range of diseases, and further studies are needed using data from patients who do not generally visit an outpatient vertigo clinic. The results in this study may be useful in terms of reflecting the actual clinical picture, but less useful in terms of interpretation of physiologic effects. Fifthly, a regular bisection task was not performed due to the time limitation, which should be addressed in further studies for comparison. The results of the deviations of the figure viewed from the center can be considered as an evaluation of a regular LBT.

## 5. Conclusions

Vestibular dysfunction can alter the deviation in the new LBT, suggesting the potential of the new LBT as an assessment of vestibular dysfunction. Both left and right vestibular dysfunctions tended to contribute to leftward deviation, suggesting that cerebral spatial perception can affect the LBT more than left or right vestibular function.

## Figures and Tables

**Figure 1 audiolres-15-00086-f001:**
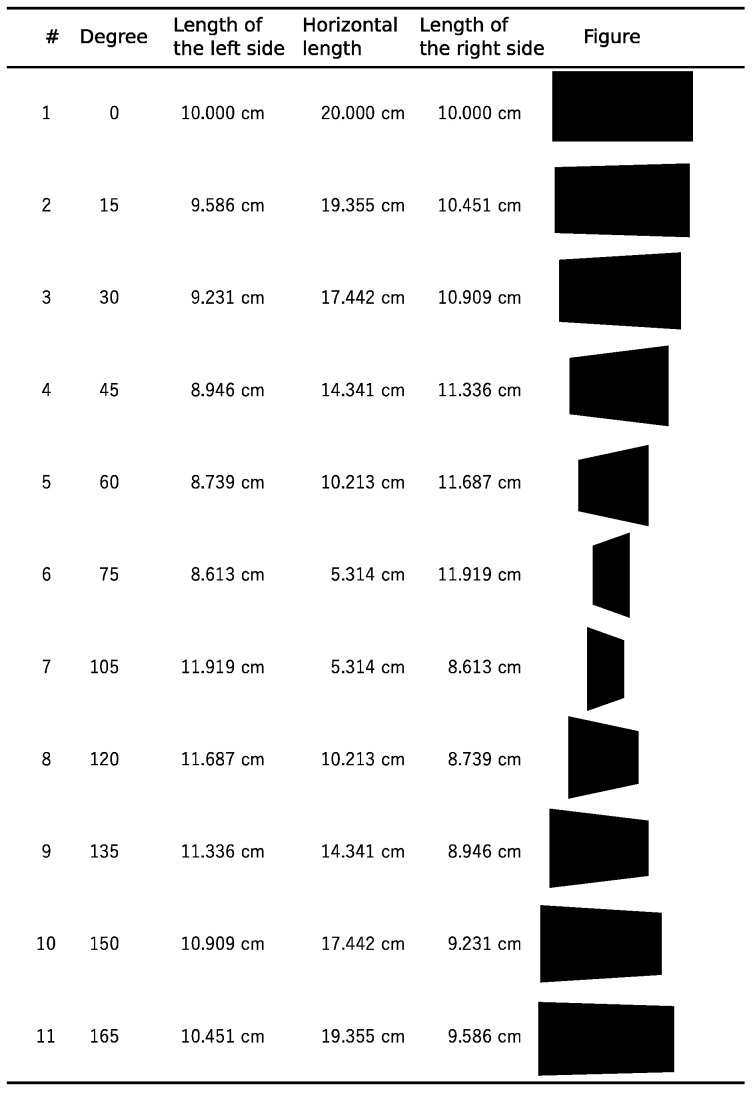
Figures used in the LBT. Figures presented in the LBT are rectangles and trapeziums, which are three-dimensionally equivalent to rectangles viewed from the side with specific degrees.

**Table 1 audiolres-15-00086-t001:** Case characteristics. The average (standard deviation) of each case characteristic for all cases. In cases of peripheral vestibular dysfunction, the caloric testing is abnormal.

Characteristics				Diagnosis		Number of Cases
Sex			*n* = 100	Functional (psychogenic) dizziness		22
	Male	53		Vestibular migraine		5
	Female	47		Orthostatic dysregulation		1
Age		54.6 (16.3)	*n* = 100	Benign paroxysmal positional vertigo (BPPV)		24
				Meniere’s disease		
DHI		28.7 (22.0)	*n* = 99		Definite	11
HADS	-A	6.7 (4.0)	*n* = 99		Probable	18
	-D	6.8 (3.6)	*n* = 99	Ramsay Hunt syndrome		1
SDS		42.0 (7.4)	*n* = 71	Vestibular neuritis		7
GDS		5.7 (4.9)	*n* = 28	Peripheral vestibular dysfunction		6
				Idiopathic bilateral vestibulopathy (IBV)		1
POUNDing		1.1 (1.3)	*n* = 99	Sensory ataxia		1
MIDAS		1.2 (0.6)	*n* = 99	Cervical hernia		1
Migraine screener			*n* = 99	Cerebellar disorders		1
	Positive	18		Normal		1
	Negative	81			Total	100

**Table 2 audiolres-15-00086-t002:** MANOVA analysis of factors of deviations of the LBT. MANOVA analysis of 0, 15, 30, 45, 60, 75, 105, 120, 135, 150, and 165 degrees: caloric testing, sex, and age.

	Df	Pillai’s Trace	Approx F	Num Df	Den Df	*p*-Value
Caloric testing	2	0.42349	2.1002	22	172	0.00442 **
Sex	1	0.13891	1.2465	11	85	0.26993
Age	1	0.20648	2.0107	11	85	0.03700 *

*: *p* < 0.05; **: *p* < 0.01; Df: degrees of freedom; approx F: MANOVA approximate F statistics; num Df: numerator degrees of freedom; den Df: denominator degrees of freedom.

**Table 3 audiolres-15-00086-t003:** Correlations between deviations of the LBT and age. The Pearson product–moment correlation coefficients and *p*-values between deviations of the LBT and age showed significant correlations in figures of 0, 15, 45, 60 and 75 degrees.

#	Degree	Age	*p*-Value
1	0	0.3924	0.0001 *
2	15	0.3114	0.0016 *
3	30	0.1817	0.0704
4	45	0.2673	0.0072 *
5	60	0.2219	0.0265 *
6	75	0.2682	0.0070 *
7	105	−0.1087	0.2818
8	120	−0.0626	0.5364
9	135	−0.0259	0.798
10	150	0.0737	0.4661
11	165	0.187	0.0624

Note: *: *p* < 0.05.

**Table 4 audiolres-15-00086-t004:** Univariate analysis of deviations of the LBT and caloric testing. The average (standard deviation) of deviations of each task and the task time in the group with canal paresis (CP) or without CP in caloric testing.

#	Degree	Caloric Testing		
		Without CP	With CP	*p*-Value
		*n* = 65	*n* = 34	
1	0	0.012 (0.215)	−0.036 (0.247)	0.313
2	15	−0.212 (0.257)	−0.117 (0.339)	0.122
3	30	−0.315 (0.284)	−0.340 (0.453)	0.741
4	45	−0.419 (0.288)	−0.308 (0.348)	0.094
5	60	−0.335 (0.269)	−0.310 (0.338)	0.687
6	75	−0.187 (0.185)	−0.138 (0.208)	0.227
7	105	0.214 (0.144)	0.213 (0.151)	0.98
8	120	0.365 (0.229)	0.344 (0.243)	0.675
9	135	0.391 (0.235)	0.273 (0.295)	0.034 *
10	150	0.309 (0.277)	0.250 (0.305)	0.335
11	165	0.135 (0.265)	0.148 (0.259)	0.81
Task time (second)		189.48 (104.27)	199.74 (99.46)	0.64

Note: *: *p* < 0.05.

**Table 5 audiolres-15-00086-t005:** Deviations in the LBT and caloric testing. The average (standard deviation) of deviations of each task in the group with left or right canal paresis (CP) or without CP in caloric testing.

#	Degree	Caloric Testing			
		Without CP	With Left CP	With Right CP	With Both CP
		*n* = 65	*n* = 14	*n* = 19	*n* = 1
9	135	0.391 (0.235)	0.254 (0.314)	0.283 (0.297)	0.362 (-)

**Table 6 audiolres-15-00086-t006:** Deviations of the LBT and psychological scores. The Pearson product–moment correlation coefficients between deviations of the LBT and psychological scores.

Degree	0	15	30	45	60	75	105	120	135	150	165	Task Time	DHI	HADS -A	HADS -D	SDS	GDS	POUN Ding	MIDAS
0	1.00	0.60	0.42	0.35	0.39	0.34	0.00	0.13	0.17	0.40	0.61	0.28	−0.14	−0.18	−0.24	−0.12	−0.46	0.02	−0.08
15	0.60	1.00	0.68	0.76	0.72	0.69	−0.39	−0.23	−0.26	0.00	0.34	0.22	−0.16	−0.19	−0.32	−0.23	−0.38	−0.02	−0.11
30	0.42	0.68	1.00	0.77	0.76	0.66	−0.40	−0.41	−0.31	−0.14	0.08	0.20	−0.11	−0.07	−0.28	−0.32	−0.05	0.08	−0.07
45	0.35	0.76	0.77	1.00	0.91	0.79	−0.59	−0.55	−0.45	−0.31	0.01	0.13	−0.10	−0.09	−0.34	−0.31	−0.25	0.05	−0.16
60	0.39	0.72	0.76	0.91	1.00	0.82	−0.59	−0.53	−0.41	−0.34	−0.03	0.09	−0.11	−0.10	−0.31	−0.33	−0.26	0.05	−0.20
75	0.34	0.69	0.66	0.79	0.82	1.00	−0.53	−0.49	−0.44	−0.28	0.00	0.25	−0.05	−0.19	−0.32	−0.34	−0.33	0.12	−0.15
105	0.00	−0.39	−0.40	−0.59	−0.59	−0.53	1.00	0.83	0.65	0.59	0.43	−0.06	0.01	−0.01	0.19	0.22	−0.13	0.09	0.09
120	0.13	−0.23	−0.41	−0.55	−0.53	−0.49	0.83	1.00	0.79	0.69	0.55	−0.04	−0.02	−0.03	0.12	0.17	−0.15	−0.02	0.12
135	0.17	−0.26	−0.31	−0.45	−0.41	−0.44	0.65	0.79	1.00	0.68	0.53	−0.05	0.00	−0.04	0.14	0.11	−0.12	−0.05	0.18
150	0.40	0.00	−0.14	−0.31	−0.34	−0.28	0.59	0.69	0.68	1.00	0.78	0.09	−0.04	−0.18	0.03	0.09	−0.33	0.01	0.16
165	0.61	0.34	0.08	0.01	−0.03	0.00	0.43	0.55	0.53	0.78	1.00	0.12	−0.16	−0.20	−0.04	0.02	−0.48	−0.02	0.05
Task time	0.28	0.22	0.20	0.13	0.09	0.25	−0.06	−0.04	−0.05	0.09	0.12	1.00	0.20	−0.12	−0.09	−0.10	−0.02	0.01	−0.02
DHI	−0.14	−0.16	−0.11	−0.10	−0.11	−0.05	0.01	−0.02	0.00	−0.04	−0.16	0.20	1.00	0.37	0.23	0.30	0.33	0.05	0.26
HADS-A	−0.18	−0.19	−0.07	−0.09	−0.10	−0.19	−0.01	−0.03	−0.04	−0.18	−0.20	−0.12	0.37	1.00	0.45	0.48	0.64	0.11	0.14
HADS-D	−0.24	−0.32	−0.28	−0.34	−0.31	−0.32	0.19	0.12	0.14	0.03	−0.04	−0.09	0.23	0.45	1.00	0.50	0.66	0.10	0.25
SDS	−0.12	−0.23	−0.32	−0.31	−0.33	−0.34	0.22	0.17	0.11	0.09	0.02	−0.10	0.30	0.48	0.50	1.00	NA	0.40	0.44
GDS	−0.46	−0.38	−0.05	−0.25	−0.26	−0.33	−0.13	−0.15	−0.12	−0.33	−0.48	−0.02	0.33	0.64	0.66	NA	1.00	−0.19	NA
POUNDing	0.02	−0.02	0.08	0.05	0.05	0.12	0.09	−0.02	−0.05	0.01	−0.02	0.01	0.05	0.11	0.10	0.40	−0.19	1.00	0.34
MIDAS	−0.08	−0.11	−0.07	−0.16	−0.20	−0.15	0.09	0.12	0.18	0.16	0.05	−0.02	0.26	0.14	0.25	0.44	NA	0.34	1.00

Note: NA: not available.

## Data Availability

The data that support the findings of this study are available from the corresponding author, upon reasonable request with the permission of the research ethics committee.

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
