# Peer review of "Vestibular Dysfunction and the Leftward Deviation in the New Line Bisection Task Using Three-Dimensionally Transformed Rectangles"

_audiolres, 2025, doi:10.3390/audiolres15040086_

Round 1

Reviewer 1 Report

Comments and Suggestions for Authors

In their work "Vestibular Dysfunction and the leftward deviation in the new Line Bisection Task using Three-Dimensionally Transformed Rectangles", Kamogashira et al. investigate the role of vestibular disorders on the performance in a 3D implementation of the line bisection task. The manuscript is well-written and the statistical analysis is accurate. The authors conclude that "Vestibular dysfunction can alter the deviation in the new LBT", although no clear clinical diagnostic value can be given yet.

While certainly a very interesting new psychophysical experiment, some major points require the authors attention.

First, I would invite the authors to extend their literatur research towards the field of vestibular stimulation in neglect, where a lot of important studies have been published, and the overall role of the vestibular system for spatial 3D perception and sensorimotor action. These studies might be crucial in explaining why a new 3D LBT version was introduced in the first place. Is there any reason why a 2D pen-and-paper test might not adequatley depict vestibular spatial deficits?

Second, the authors need to adress some major methodological flaws of the current study, especially the lack of a control cohort. Spatial abilities vary widely in healthy adults, and so does LBT performance. Without a control cohort, no clear effect of vestibular (dys-) function on test performance can be proven, and this fact needs to be addressed as a limitation. If possible, a small cohort of a handful of healthy adults (e.g., authors, staff members, friends and family ...) also performing the 3D-LBT to provide a sort of normative data would greatly improve the study, although this is just a suggestion. If the experimental setup is not available anymore, I understand this.

Third, I invite the authors to extend their discussion on how both a left- and right-sided peripheral vestibular deficit might cause a left-sided deviation of the 3D-LBT. This is a very interesting finding which requires further attention. Was there any effect of the acuity of the vestibular deficit on this effect, e.g., as a sign of central-vestibular compensation strategies? If I see it correctly, no SVV testing was performed, but maybe clinical records allow for an assessment of vestibular deficit timecourse.

If I see it correctly, no correlation between 3D-LBT and any of the psychometric questionnaires (DHI, HADS, ...) was found. I think this should be at least noted in the main manuscript, and could be discussed in a paragraph. While the DHI is very unspecific, the role of anxiety in spatial performance has been commonly investigated. If there were significant effects, this should also be noted in the main manuscript.

In conclusion, I invite the authors to extend some parts of the manuscript to fully represent the current state of research, and to extend the discussion. If no control cohort can be added, this should also be discussed as a major limitation of the study.

Author Response

Reviewer 1

Comments and Suggestions for Authors

In their work "Vestibular Dysfunction and the leftward deviation in the new Line Bisection Task using Three-Dimensionally Transformed Rectangles", Kamogashira et al. investigate the role of vestibular disorders on the performance in a 3D implementation of the line bisection task. The manuscript is well-written and the statistical analysis is accurate. The authors conclude that "Vestibular dysfunction can alter the deviation in the new LBT", although no clear clinical diagnostic value can be given yet.

While certainly a very interesting new psychophysical experiment, some major points require the authors attention.

Thank you very much for the comments and suggestions. These suggestions helped us to create a better version of the article.

First, I would invite the authors to extend their literatur research towards the field of vestibular stimulation in neglect, where a lot of important studies have been published, and the overall role of the vestibular system for spatial 3D perception and sensorimotor action. These studies might be crucial in explaining why a new 3D LBT version was introduced in the first place. Is there any reason why a 2D pen-and-paper test might not adequatley depict vestibular spatial deficits?

Thank you very much for the comments.

We added several literature according to this suggestion.

There are several reports of vestibular dysfunction in PN, with many studies referring to the function of neural circuits in the cerebrum. The hemispatial neglect can develop after acute unilateral peripheral vestibulopathy, which is attributed to damaged vestibular subnuclei, which receive afferents from both peripheral vestibular end organs and the vestibulocerebellum and project to the ipsilateral or contralateral thalamus and vestibular cortex [4]. However, in another study, vestibular tone imbalance due to unilateral impairment of the vestibular organs did not cause spatial hemineglect, with mild attention deficits in both visual spaces [10].

Visuospatial neglect involves the central influence of vestibular stimuli on the mechanisms of spatial representation [7]. The bias to the right suggests that the subject on the left is being ignored, which occurs in the superior temporal cortex, insula, and temporo-parietal junction involved in the multisensory system, including vestibular function. These areas integrate multimodal functions of vestibular, auditory, cervical proprioceptive, and visual input to form higher-order spatial representations [5].

Asymmetric vestibular input to cortical areas leads to representational spatial deficits and spatial cognition, as well as deficits in cortical processing of vestibular input in spatial neglect after right hemisphere stroke [9]. In several studies on rehabilitation for post-stroke hemispatial neglect showed that vestibular stimulation may have therapeutic potential, results have been inconsistent and further studies based on more careful methodology are needed [6].

The use of a PC makes data measurement and collection much easier.

Second, the authors need to adress some major methodological flaws of the current study, especially the lack of a control cohort. Spatial abilities vary widely in healthy adults, and so does LBT performance. Without a control cohort, no clear effect of vestibular (dys-) function on test performance can be proven, and this fact needs to be addressed as a limitation. If possible, a small cohort of a handful of healthy adults (e.g., authors, staff members, friends and family ...) also performing the 3D-LBT to provide a sort of normative data would greatly improve the study, although this is just a suggestion. If the experimental setup is not available anymore, I understand this.

Thank you very much for the comments.

We presented the data of the control group of Asia University in Table A8, whereas age was found to significantly affect deviation, so direct comparisons between the control group of Asia University and the outpatient group were not made because the age distribution is very different from that of outpatient cases.

Third, I invite the authors to extend their discussion on how both a left- and right-sided peripheral vestibular deficit might cause a left-sided deviation of the 3D-LBT. This is a very interesting finding which requires further attention. Was there any effect of the acuity of the vestibular deficit on this effect, e.g., as a sign of central-vestibular compensation strategies? If I see it correctly, no SVV testing was performed, but maybe clinical records allow for an assessment of vestibular deficit timecourse.

Thank you very much for the comments. It was predicted that which of the cerebral hemispheres was activated was a more important factor than the left-right difference of the vestibular dysfunction. We added the following sentence regarding psychological scores.

No significant correlation was found between the task time or deviations in the LBT and questionnaire scores regarding psychological symptoms in this study. One study of stroke patients found a significant association between the LBT time and prognosis of hemispatial neglect, indicating that cognitive deficits such as attention and visual perception affect recovery of hemispatial neglect [47]. In a study examining the LBT in normal subjects, generalized anxiety disorder (GAD), and treatment-resistant depression (TRD) outpatients, a significant leftward deviation was observed in TRD patients compared to normal subjects [48]. In another study, patients with dependent personality disorder bisected significantly leftward compared to healthy controls [49]. In the LBT study on headache, normal subjects and migraine patients showed the rightward deviation, while tension-type headache patients, in whom psychological stress is mainly involved, showed the leftward deviation compared to normal subjects and migraine patients, suggesting that right hemisphere activation may be relatively stronger or left hemisphere activation weaker in tension-type headache [50]. While the control group of Asia University showed a significant leftward deviation in 0 degree, the outpatients in this study showed no significant deviation in 0 degree, which could be considered the opposite results of previous studies above, given that patients tend to show more depression than the control group of Asia University. The age factor may contribute more strongly to the deviation in the LBT than the psychogenic factor. In addition, a leftward deviation was observed in vestibular dysfunction regardless of left-side or right-side, suggesting that the impaired side of the vestibule was less affected.

In terms of approach motivation, the significant rightward deviation was observed in challenge condition with high self-esteem, indicating the effectiveness of LBT in measuring behavioral neuroscience indicators [45]. In another study, approach-motivated individuals exhibited a significant rightward deviation compared to avoidance-motivated individuals, but only under conditions of high time pressure [46]. However, other research groups have argued that the LBT was not suitable for measuring lateralized approach/avoidance biases, probably because of the large individual differences inherent in visuospatial bias [47].

If I see it correctly, no correlation between 3D-LBT and any of the psychometric questionnaires (DHI, HADS, ...) was found. I think this should be at least noted in the main manuscript, and could be discussed in a paragraph. While the DHI is very unspecific, the role of anxiety in spatial performance has been commonly investigated. If there were significant effects, this should also be noted in the main manuscript.

Thank you very much for the comments. We moved the table into the main manuscript and added a paragraph discussing this issue.

In conclusion, I invite the authors to extend some parts of the manuscript to fully represent the current state of research, and to extend the discussion. If no control cohort can be added, this should also be discussed as a major limitation of the study.

Thank you very much for the comments. We added the following sentence.

Both left and right vestibular dysfunctions tended to contribute to the leftward deviation, suggesting that cerebral spatial perception can affect the LBT more than left or right vestibular function.

Since this study was not conducted simultaneously due to time constraints, a comparison with classical LBT needs to be additionally studied in the future and is listed as a limitation.

Fifthly, a regular bisection task was not enforced due to the time limitation, which should be addressed in further studies for comparison.

Reviewer 2 Report

Comments and Suggestions for Authors

Tile

Vestibular Dysfunction and the leftward deviation in the new Line Bisection Task using Three-Dimensionally Transformed Rectangles

This is an interesting and novel study, that explore the utility of a popular neuro test of spatial neglect in vestibular assessment. Although the findings are not really substantial as such, this valuable preliminary information indicates need of further exploration in this line. They found that presence of vestibular dysfunction may result in variations in LBT performance.

The introduction is brief and comprehensive. However, for the improved readability slight modifications in the structure (mainly dividing content into paragraphs based on topic of discussion) could be done.

Page 2 line 46

I feel it is better to introduce the role of vestibular function on PN as a new paragraph.

Line 68-69

It is mentioned that new LBT with 3d rectangles are used for enhanced depth perception. Is there any specific evidence or logic for this? Will enhanced depth perception improve sensitivity of LBT? Or the new modification is somehow helping efficacy of LBT in vestibular assessment? I feel including such information strengthen justification for the new version of LBT.

Line 69

I understand that the conventional LBT uses lines for the bisection task. Here you developed 3D rectangular shapes, i.e. the subject has to bisect the rectangle, not line. So,  Why did you still name it LBT or new LBT? Why can’t you give a different name, something that indicates the difference in the stimulus but maintaining term that indicates bisection task ?

The purpose of the study is stated as “to evaluate the influence of vestibular function on the deviation of the new LBT with three-dimensionally transformed rectangles in cases with

vestibular dysfunction”. Providing more information about specific objectives of the study would enhance the clarity of the study. As one read further through the result section, there is a chance of feeling gap between broadly stated purpose/ aim of the study and detailed analysis mentioned. For example,  regression analyses were done, comparison between different vhit, vemp patterns were done. In addition, how control group is utilized also not clear. Elaborating study objectives would make these analyses more graspable.

Results

Appreciate that the authors provided statistical data as tables within main content as well as in appendices. However, division of tables between main content and appendices is not clear. What is the basis of the divisions of tables? Why can’t tables with descriptive statistics be retained within main content and tables with statistical test results moved to appendix (with relevant statistical values mentioned in the text itself).

In addition, there appears an inconsistency in how tables in appendix is mentioned. Somewhere it is mentioned as supplementary table 8, while others it is mentioned with proper numbering like Table A1 or A2. Consider using uniform citations of tables through out the manuscript. 

Table 1: consider adding clear column titles (specifically the right part where the number of different diagnoses is presented).

Table 3: The column title (3rd column) could be ‘age’, it looks confusing. Is it coefficient correlation values are presented?

Similarly, check all the tables and consider adding/modifying column titles and subtitles wherever possible to maximize the comprehensibility of data.

Discussion

Well written. Mentioned the limitations and future directions

Conclusion

The conclusion section  could also be elaborated briefly for better clarity. Merely a statement like ‘vestibular dysfunction affects lbt perfromance’ may be too general to fully convey the study’s findings. may be authors can improve this section after modifying the aim/objectives section and make it in align with the objectives.

Author Response

Reviewer 2

Comments and Suggestions for Authors

Tile

Vestibular Dysfunction and the leftward deviation in the new Line Bisection Task using Three-Dimensionally Transformed Rectangles

This is an interesting and novel study, that explore the utility of a popular neuro test of spatial neglect in vestibular assessment. Although the findings are not really substantial as such, this valuable preliminary information indicates need of further exploration in this line. They found that presence of vestibular dysfunction may result in variations in LBT performance.

Thank you very much for the comments and suggestions. These suggestions helped us to create a better version of the article.

The introduction is brief and comprehensive. However, for the improved readability slight modifications in the structure (mainly dividing content into paragraphs based on topic of discussion) could be done.

Page 2 line 46

I feel it is better to introduce the role of vestibular function on PN as a new paragraph.

Thank you very much for the comments. The similar concern is raised by another reviewer. We added several previous studies.

There are several reports of vestibular dysfunction in PN, with many studies referring to the function of neural circuits in the cerebrum. The hemispatial neglect can develop after acute unilateral peripheral vestibulopathy, which is attributed to damaged vestibular subnuclei, which receive afferents from both peripheral vestibular end organs and the vestibulocerebellum and project to the ipsilateral or contralateral thalamus and vestibular cortex [4]. However, in another study, vestibular tone imbalance due to unilateral impairment of the vestibular organs did not cause spatial hemineglect, with mild attention deficits in both visual spaces [10].

Visuospatial neglect involves the central influence of vestibular stimuli on the mechanisms of spatial representation [7]. The bias to the right suggests that the subject on the left is being ignored, which occurs in the superior temporal cortex, insula, and temporo-parietal junction involved in the multisensory system, including vestibular function. These areas integrate multimodal functions of vestibular, auditory, cervical proprioceptive, and visual input to form higher-order spatial representations [5].

Asymmetric vestibular input to cortical areas leads to representational spatial deficits and spatial cognition, as well as deficits in cortical processing of vestibular input in spatial neglect after right hemisphere stroke [9]. In several studies on rehabilitation for post-stroke hemispatial neglect showed that vestibular stimulation may have therapeutic potential, results have been inconsistent and further studies based on more careful methodology are needed [6].

Line 68-69

It is mentioned that new LBT with 3d rectangles are used for enhanced depth perception. Is there any specific evidence or logic for this? Will enhanced depth perception improve sensitivity of LBT? Or the new modification is somehow helping efficacy of LBT in vestibular assessment? I feel including such information strengthen justification for the new version of LBT.

Thank you very much for the comments. We added several previous studies.

Several previous studies have reported the importance of depth perception in the LBT, and our new LBT uses deformation to represent depth. Depth perception is influenced by left-right alignment along the horizontal axis [19] and visuospatial reality construction through virtual reality was associated with a rightward LBT bias in the virtual environment [20]. The preference for rectangularity is stronger than that for symmetry [17], and rectangularity is effective for shape perception by incorporating some additional constraints [18].

Line 69

I understand that the conventional LBT uses lines for the bisection task. Here you developed 3D rectangular shapes, i.e. the subject has to bisect the rectangle, not line. So, why did you still name it LBT or new LBT? Why can’t you give a different name, something that indicates the difference in the stimulus but maintaining term that indicates bisection task ?

Since this study was not conducted simultaneously due to time constraints, a comparison with classical LBT needs to be additionally studied in the future and is listed as a limitation.

Fifthly, a regular bisection task was not enforced due to the time limitation, which should be addressed in further studies for comparison.

The purpose of the study is stated as “to evaluate the influence of vestibular function on the deviation of the new LBT with three-dimensionally transformed rectangles in cases with vestibular dysfunction”. Providing more information about specific objectives of the study would enhance the clarity of the study. As one read further through the result section, there is a chance of feeling gap between broadly stated purpose/ aim of the study and detailed analysis mentioned. For example, regression analyses were done, comparison between different vhit, vemp patterns were done. In addition, how control group is utilized also not clear. Elaborating study objectives would make these analyses more graspable.

Thank you very much for the comments. Please refer to Table 1 describing the case characteristics. The details of each statistical analysis are described in the figure descriptions. Similar concerns were raised by another reviewer. We presented the data of the control group of Asia University in Table A8, whereas age was found to significantly affect deviation, so direct comparisons between the control group of Asia University and the outpatient group were not made because the age distribution is very different from that of outpatient cases.

Results

Appreciate that the authors provided statistical data as tables within main content as well as in appendices. However, division of tables between main content and appendices is not clear. What is the basis of the divisions of tables? Why can’t tables with descriptive statistics be retained within main content and tables with statistical test results moved to appendix (with relevant statistical values mentioned in the text itself).

Thank you very much for the comments.

Data on age and vestibular disorders that were statistically significant were selected for the tables in the text. Other parameters for which there were no statistically significant differences were basically presented in the appendices.

In addition, there appears an inconsistency in how tables in appendix is mentioned. Somewhere it is mentioned as supplementary table 8, while others it is mentioned with proper numbering like Table A1 or A2. Consider using uniform citations of tables throughout the manuscript.

Thank you very much for the comments. We fixed the numbering.

Table 1: consider adding clear column titles (specifically the right part where the number of different diagnoses is presented).

Thank you very much for the comments. We fixed the title.

Table 3: The column title (3rd column) could be ‘age’, it looks confusing. Is it coefficient correlation values are presented?

Thank you very much for the comments. We described the statistical analysis in the legend as follows.

Correlations between deviations of the line bisection task and age. The Pearson product-moment correlation coefficients and p-values between deviations of the line bisection task and age showed significant correlations in figures of 0, 15, 45, 60 and 75 degrees.

Similarly, check all the tables and consider adding/modifying column titles and subtitles wherever possible to maximize the comprehensibility of data.

Thank you very much for the comments. We checked the tables thoroughly.

Discussion

Well written. Mentioned the limitations and future directions

Thank you very much for the comments and suggestions.

Conclusion

The conclusion section could also be elaborated briefly for better clarity. Merely a statement like ‘vestibular dysfunction affects lbt perfromance’ may be too general to fully convey the study’s findings. may be authors can improve this section after modifying the aim/objectives section and make it in align with the objectives.

Thank you very much for the comments. Similar concerns were raised by another reviewer. We added the following sentence.

Both left and right vestibular dysfunctions tended to contribute to the leftward deviation, suggesting that cerebral spatial perception can affect the LBT more than left or right vestibular function.

Round 2

Reviewer 1 Report

Comments and Suggestions for Authors

All of my concerns have been addressed.